# Effect of Chickpea Dietary Fiber on the Emulsion Gel Properties of Pork Myofibrillar Protein

**DOI:** 10.3390/foods12132597

**Published:** 2023-07-04

**Authors:** Dianbo Zhao, Shuliang Yan, Jialei Liu, Xi Jiang, Junguang Li, Yuntao Wang, Jiansheng Zhao, Yanhong Bai

**Affiliations:** 1College of Food and Bioengineering, Zhengzhou University of Light Industry, Zhengzhou 450001, China; zhaodb212@163.com (D.Z.); yanshuliang2023@163.com (S.Y.); 18211723136@163.com (J.L.); jxx202306@163.com (X.J.); 2014081@zzuli.edu.com (J.L.); 2017019@zzuli.edu.cn (Y.W.); 2Henan Key Laboratory of Cold Chain Food Quality and Safety Control, Key Laboratory of Cold Chain Food Processing and Safety Control, Ministry of Education, Henan Collaborative Innovation Center for Food Production and Safety, Zhengzhou University of Light Industry, Zhengzhou 450001, China; 3Henan Food Laboratory of Zhongyuan, Zhengzhou University of Light Industry, Luohe 462000, China; zjs4567@163.com

**Keywords:** chickpea dietary fiber, myofibrillar protein, emulsion gel, gel property

## Abstract

In this study, the effect of chickpea dietary fiber (CDF) concentration (0%, 0.4%, 0.8%, 1.2%, 1.6%, and 2.0%) on emulsion gel properties of myofibrillar protein (MP) was investigated. It was found that the emulsifying activity index (EAI) and emulsifying stability index (ESI) of MP increased with the increasing content of CDF. Moreover, the water- and fat-binding capacity (WFB), gel strength, storage modulus (G’), and loss modulus (G”) of MP emulsion gel also increased with increasing content of CDF. When the concentration of CDF was 2%, the most significant improvement was observed for EAI, breaking force, and WFB (*p* < 0.05); the three-dimensional gel network structure of the MP emulsion gel was denser and the pore diameter was smaller. The T_21_ relaxation time of emulsion gel decreased while the PT_21_ increased significantly with the increasing content of CDF, suggesting that the emulsion gel with CDF had a better three-dimension network. The addition of CDF led to an increased content of *β*-sheet and reactive sulfhydryl and increased surface hydrophobicity of MP, thus improving the gel properties of the MP emulsion gel. In conclusion, the addition of CDF improved the functional properties and facilitated the gelation of the MP emulsion, indicating that CDF has the potential to improve the quality of emulsified meat products.

## 1. Introduction 

Myofibrillar protein (MP), an important functional protein in muscle, is composed of myosin, actomyosin, actin, tropomyosin, troponin, and other components, accounting for about 55~65% of the muscle protein [1]. Moreover, MP has an important influence on the quality and functional characteristics of meat products [2]. In the processing of meat products, MP played an important role in the stabilization of water and fat. For example, in the formulation of meat products such as Frankfurters and Bologna sausages, the fat is cut into small spheres and stabilized by meat proteins, so they can be classified as protein-based emulsion gels, which is a kind of gel obtained by the gelation of protein-stabilized emulsion [3]. MP emulsion gels formed by heating contributed greatly to the texture and cohesion of the meat products [4]. Therefore, improving the functional properties of MP may result in meat products with desirable texture [5]. While MP emulsion gel is mainly dependent on the formation of the three-dimensional network structure of MP in the aqueous phase, and its structural characteristics are key factors affecting the properties of emulsion gel [6]. However, the factors which affect the structural characteristics of proteins, such as pH, ionic strength, etc., can affect the stability of emulsion gel [7], so emulsion stabilized by protein matrix alone has poor stability. 

To improve the quality of MP emulsion gel, domestic and foreign researchers are gradually developing new additives for meat products, such as soy proteins [8] and konjac [9], which can provide essential nutrients for our dairy life and play a key role in the formation of textures [10]. Dietary fiber (DF) is an indigestible carbohydrate, but its intake has numerous benefits, including increasing intestinal volume and transport, regulating blood sugar levels, and so on [11]. At present, dietary fiber has been widely reported as a functional ingredient in processed meat products [12]. Many attempts have been made to improve the water-holding capacity, emulsifying properties, and gel properties of meat products by adding dietary fiber. Zhuang et al. [13] studied the effect of sugarcane dietary fiber on the physical and chemical properties of low-fat sausage and found that the addition of insoluble dietary fiber significantly improved the textural properties of the final product. Pinero et al. [14] made a low-fat beef burger with 13.45% oat fiber and found that the meat pie juice retained was more than in the control group, which may be attributed to the good water retention capacity of oat fiber, reducing water loss during baking. Debusca, Tahergorabi, Beamer, Matak, and Jaczynski [15] found that the properties of fish MP gel could be improved by adding different amounts of dietary fiber (0~8%). When the dosage reached 6%, the texture and color of the gel reached the maximum.

Chickpea (*Cicer arietinum* L.), an excellent source of protein and carbohydrates, is the third most cultivated bean plant in the world, which has attracted much attention in recent years due to its potential nutritional and health advantages such as antioxidant, hypoglycemic, and antitumor activity [16,17]. Moreover, chickpea dietary fiber (CDF) contains a large amount of dietary fiber (14~26%) [18] and a relatively small amount of fat (5.5~6.9%) [19]. At present, most scholars are focused on the physicochemical and functional properties of chickpea dietary fiber [20], while the potential mechanism of CDF that affects the emulsion gel properties of proteins has been rarely reported. In this study, CDF was added to pork MP as a dietary fiber to study its emulsion gel properties by measuring gel strength, water- and fat-binding capacity (WFB), and rheological properties. The interaction between CDF and MP was further studied by measuring the secondary structure, active sulfhydryl group (R-SH), and surface hydrophobicity of MP. Evaluating the improved functional properties of MP via the addition of CDF has guiding significance for the formulation of meat products.

## 2. Materials and Methods

### 2.1. Materials

Porcine longissimus dorsi was purchased from a local supermarket (Zhengzhou, Henan, China). The meat was trimmed of any excess fat and obvious connective tissue and then stored at −20 °C until required for extraction of MP. Chickpeas (*Cicer arietinum* L.) were obtained from Yunnan province, China. All other reagents and chemicals used were of analytical grade, provided by Aladdin Chemical Reagent Co., Ltd. (Shanghai, China).

### 2.2. Extraction of Myofibrillar Proteins (MP)

MP was prepared as previously reported with some modifications [21]. Briefly, the longissimus dorsi of pork was quickly cut into small pieces and then ground two times at 2000 r/min for 10 s through a Waring Blender (GM200, Restch, Haan, Germany). MP was extracted with four volumes of cooling isolation buffer (10 mM Na_2_HPO_4_/NaH_2_PO_4_, 0.1 M NaCl, 2 mM MgCl_2_, 1 mM EGTA, pH 7.0, 4 °C) and centrifugated (Beckman L-90K, Beckman, Pasadena, CA, USA) at 3000× *g* for 15 min at 4 °C. The supernatant was poured out, and the pellets were re-suspended with a homogenizer (Ultra Turrax T-25 BASIC, IKA Company, Staufen, Germany) and centrifuged twice more under the same conditions. Then, the pellet was mixed in four volumes of salt solution (0.1 M NaCl), centrifuged (3000× *g* for 15 min), and washed three times. Before the final wash, the MP was filtered with three layers of gauze. The final pellet collected was MP, stored at 4 °C for further use within 48 h.

### 2.3. Preparation of Dietary Fiber from Chickpea

As the fat in chickpea will influence the later experiments, the defatted chickpea flour was obtained in laboratory according to a previous method [22]. CDF was produced from the defatted chickpea flour according to a previously reported method with minor modifications [23]. A total of 100 grams of defatted chickpea powder was soaked in 1.5 L 0.5 M NaOH at 50 °C for 2 h. Next, the supernatants were removed by centrifugation (3000× *g* for 15 min) and the precipitates were washed with 0.5 M NaOH and then centrifuged. The precipitates were suspended in 1 L of 1.0 M HCl at 50 °C for 2 h. After centrifuging (3000× *g* for 15 min), the soluble fractions were removed then the precipitates were washed with 0.5 M HCl and neutralized with alkali. Finally, the precipitates were washed twice with distilled water and freeze dried (LyoQuest-85, Telstar, Barcelona, Spanish). CDF with a yield of 10.8% was obtained and ground into powder using a grinder (LK-400A, Xinnuo, Shanghai, China), then passed through a 100-mesh sieve (100-mesh, Hongmei, Yangzhou, Jiangsu, China). 

### 2.4. Determination of Emulsifying Activity Index (EAI) and Emulsifying Stability Index (ESI) 

The MP was diluted with 0.6 M NaCl buffer (0.6 M NaCl, 20 mM Na_2_HPO_4_/NaH_2_PO_4_, pH 7.0) to 8 mg/mL. Six groups of MP with CDF content of 0%, 0.4%, 0.8%, 1.2%, 1.6%, and 2.0% were prepared. Next, the MP-CDF solution and soybean oil (4:1, *w*/*v*) were mixed and homogenized three times at 10,000 r/min for 20 s with a homogenizer (Ultra Turrax T-25 BASIC, IKA Company, Staufen, Germany). Then, the emulsifying properties of the MP-CDF composite were evaluated. The storage stability of the emulsions was recorded at 4 °C for 0 h, 24 h, 72 h, and 120 h, respectively. The EAI and ESI of MP were measured according to the method of Shen [24]. The fresh emulsion was immediately collected from the bottom of the tube and mixed with 5 mL SDS solution (1 mg/mL), After vortexing for 30 s, the absorbance was determined at 500 nm using a UV/VIS spectrometer (TU-1810, PERSEE, China). Ten minutes later, the absorbance was determined by taking 0.05 mL emulsion with the same method. The EAI was calculated using the formula as below:(1)$$\mathrm{EAI}\;(m^{2}/g)=\frac{2.303\times 2\times A_{0}\times N}{c\times \Phi \times 10^{4}}$$
(2)$$\mathrm{ESI}\;(\min)=\frac{10\times A_{0}}{A_{0}-A_{10}}$$
where *A*_0_ and *A*_10_ are the absorbance values at 0 min and 10 min after the formation of emulsion, respectively; *N* is the dilution factor; *c* is the protein concentration (mg/mL); and Φ is the volume fraction of soybean oil. 

### 2.5. Storage Stability of the Emulsion

The effect of different concentrations of CDF on storage stability of MP emulsion were investigated by determining the height of the emulsion layer after storage for a different time.

### 2.6. Preparation of Emulsion Gel

MP was diluted with 0.6 M NaCl buffer to ensure the final protein concentration was 30 mg/mL. Next, CDF (0, 0.4, 0.8, 1.2, 1.6, and 2 g/100 g) was added and the mixture was homogenized (Ultra Turrax T-25 BASIC, IKA Company, Staufen, Germany) for 60 s at 10,000 r/min to ensure homogeneity of the sample. Soybean oil was added to the MP-CDF solutions (1:4, *v*/*w*) and homogenized three times at 10,000 r/min for 20 s. Part of the samples were taken to evaluate the dynamic rheological properties; then, the rest of the samples were centrifuged at 800× *g* for 5 min to remove air bubbles before heating in a water bath at 80 °C for 30 min [25]. Then, the samples were stored at 4 °C overnight. Prior to measurement, the gels were equilibrated at room temperature (25 °C) for 30 min.

### 2.7. Water- and Fat-Binding (WFB) Capacity

The WFB of the samples was measured based on the method of Zhuang [3]. The heated MP gels were centrifuged at 10,000× *g* for 15 min at 4 °C and the supernatant was removed; then, the weight of the gel before and after centrifugation was recorded. WFB was calculated using the following equation:(3)$$\mathrm{WFB}\;(\%)=\frac{m_{1}-m_{0}}{m_{2}-m_{0}}\times 100\%$$
where *m*_1_ is the weight of the tube and gel after centrifugation, g; *m*_2_ is the weight of tube and gel before centrifugation, g; and *m*_0_ is the weight of centrifuge tube, g.

### 2.8. Breaking Force (BF)

The BF of the samples was measured according to the method of Li [26] with some modifications. The gel samples were equilibrated at room temperature for 30 min and then cut up into cylinders (20 mm in height and diameter). Next, the samples were measured using a texture analyzer (TA-XT Plus, Stable Micro Systems, London, UK) with a P/0.5 plate probe. The parameters for testing were as follows: pre-test speed 2.0 mm/s, text speed 1.0 mm/s, post-test speed 2.0 mm/s, distance 10 mm, and trigger force 5 g. The peak force needed to breach the gel was described as the breaking force. Each sample was analyzed five times. 

### 2.9. Low-Field NMR

The low-field NMR of the samples was performed as previously described by Han with some modifications [21]. The samples were measured with an NMR analyzer (NMI120, Niumag Electric Corp., Suzhou, Jiangsu, China) operating at 18 MHz. The *T*_2_ was measured using the CPMG with 32 scans, 12,000 echoes, 250 μs, between pulses of 90° and 180°. The repetition time between subsequent scans was 2000 ms. The low-field NMR relaxation curves (*T*_2_) were analyzed with the MultiExp Inv Analysis software (Niumag Electric Corp., Suzhou, Jiangsu, China). 

### 2.10. Dynamic Rheological Measurements 

The dynamic rheological (frequency sweep and temperature sweep) properties of MP-CDF emulsion were evaluated using a rheometer (Discovery HR-1, TA Instruments, New Castle, USA). The sample was loaded between 40 mm diameter parallel plates and the gap was 0.4 mm.

#### 2.10.1. Frequency Sweep Test

The frequency sweep tests were measured from 0.1 rad/s to 100 rad/s at 25 °C and the strain was set as 1.0%, which was within the linear viscoelastic region. The dynamic storage modulus (G’) and loss modulus (G”) were recorded.

#### 2.10.2. Temperature Sweep Measurements

The samples were sealed with silica oil to prevent the evaporation of water during the test. The parameters of the test were set as follows: the frequency was 0.1 Hz, the strain was 1.0%, and the sample was heated from 20 °C to 80 °C at a heating rate of 2 °C/min.

### 2.11. Cryogenic Scanning Electron Microscope (Cryo-SEM)

The gels were observed with a Cryo-SEM (SU8010, Hitachi, Tokyo, Japan) at an accelerating voltage of 1.0 kV. The emulsion gel sample was fixed on the sample stage with glue and inserted into liquid nitrogen for rapid freezing, then transferred to the sample platform for observation of the microstructure.

### 2.12. Secondary Structure

Ultraviolet circular dichroism (UV CD) spectroscopy was performed with a Chriascan spectrometer (Applied Photo Physics, Surrey, UK) to evaluate the effect of CDF on the secondary structure of MP (0.2 mg/mL) according to a previous method [6]. MP was scanned from 190 nm to 260 nm, and scan rate, optical path, and band width were set as 100 nm/min, 0.1 cm, and 1 nm, respectively. The secondary structure of MP was calculated using the CDNN software(v2.1, Applied Photophysics Ltd, London, UK).

### 2.13. Surface Hydrophobicity

The surface hydrophobicity of MP-CDF was determined according to the procedures of Li [27]. Firstly, the concentration of MP was adjusted to 2.0 mg/mL with 0.6 M NaCl buffer. Then, MP (2 mL) was mixed with 80 μL bromophenol blue (1 mg/mL) (BPB), while the control group was prepared by mixing 0.6 M NaCl buffer with 80 μL BPB. All the samples were kept at 20 °C for 10 min and then centrifuged at 4000× *g* for 15 min. After centrifugation, the supernatants of the sample and the control were diluted ten times; then, the absorbance was measured at 595 nm. The content of the BPB bond was calculated using the following equation: (4)$$\mathrm{BPB}\;\mathrm{bound}\;(\mu g)=\frac{80\times \left(A_{0}-A_{S}\right)}{A_{0}}$$
where *A*_0_ and *A*_s_ are the absorbance of the control and the sample, respectively. 

### 2.14. Reactive Sulfhydryl (R-SH)

The R-SH of the sample was calculated based on the method of Guo [28]. MP-CDF suspensions were diluted to 4.0 mg/mL with 0.6 M NaCl buffer. Next, 50 μL of 5,5’-dithiobis-2-nitrobenzoic acid (10 mM DTNB in 0.1 M phosphate buffer, pH 8.0) was added to 5 mL solution and incubated in dark at 4 °C for 1 h. Then, the absorbance was measured at 412 nm. The R-SH content was calculated using the molar extinction coefficient of 13,600 M^−1^cm^−1^.

### 2.15. Statistical Analysis

Three parallel trials were carried out in this experiment, and the data are expressed as the mean ± standard errors. Data were analyzed using IBM SPSS software (version 20.0, IBM, Chicago, IL, USA) and Origin 9.0, and the differences among treatments were determined by a significance level of *p* < 0.05.

## 3. Results and Discussion

### 3.1. Storage Stability of Emulsions

The emulsifying properties of proteins had a close relationship with the properties of the resulting protein emulsion gel [29], so the effect of CDF on the properties of emulsion stabilized by MP was studied first. The effect of different concentrations of CDF on storage stability of MP emulsion was investigated by determining the height of the emulsion layer. Usually, emulsions with a higher emulsion layer had better storage stability [30]. Figure 1 shows the storage stability of the MP-CDF emulsion after storage at 4 °C for 0 h, 24 h, 72 h, and 120 h. Compared with the control group, the emulsion with CDF had better stability, and the concentration of CDF had a significant influence on the emulsion stability. The control group showed obvious stratification after 24 h, while in the treatment group, no stratification occurred except for slight stratification when 0.4% CDF was added. Moreover, no stratifications were observed at 24, 72, or 120 h for the group with CDF contents of 1.6% and 2.0%. These results indicated that MP emulsion with different CDF concentrations had a better storage stability, and the stability of emulsion gradually enhanced with the increasing concentration of CDF. This was probably because the presence of CDF enhanced the steric hindrance effect and increased the viscosity of the emulsion. Xu [31] also reported that the addition of polysaccharides increased the viscosity of the emulsion, decreased the movement rate of the particles in the emulsion, and prevented flocculation and aggregation of the emulsion.

### 3.2. EAI and ESI

Emulsion is a mixture of two or more immiscible liquids. EAI and ESI are important indices to evaluate the emulsifying characteristics of samples [32]. The effect of CDF on the emulsifying properties of MP is shown in Figure 2. With the increasing content of CDF, the EAI and ESI of the MP emulsion showed a significant increase (*p* < 0.05). The EAI of MP-CDF increased from 22.22 m^2^/g to 38.35 m^2^/g after adding 2% CDF. This may be associated with the fact that MP and CDF could be adsorbed on the surface of oil droplets together, which increased the interfacial film thickness at the oil–water interface (Figure 2) and was conducive to the improvement in EAI. It was reported that proteins could be adsorbed and form a protective film on the surface of oil droplets through hydrophobic force, maintaining the stability of the emulsion system [33]. Thus, the control MP showed a certain emulsifying stability and the EAI was 91.73%. The ESI of the MP-CDF emulsion system were significantly higher than that of the control group after the addition of CDF. The ESI of CDF-MP increased by 69.70% compared with the control when the concentration of CDF was 2.0%. This was probably because CDF had strong interaction with the hydrophobic groups in MP on the oil–water interface of the emulsion, and then inhibited protein precipitation and maintained the stability of the system. Sun and Gunasekaran [34] reported that polysaccharides were highly hydrophilic molecules which usually had a certain viscosity and helped form a hydrophilic boundary layer to stabilize the emulsion. 

### 3.3. Breaking Force (BF) Analysis

Breaking force (BF) can objectively reflect the gel-forming ability of proteins and is an important means to evaluate the final quality of emulsion gel [35]. It can be seen from Figure 3 that when 2% CDF was added, the BF of the emulsion gel increased significantly from 93.38 g to 177.59 g (*p* < 0.05). This was probably because a higher concentration of CDF could form a weak DF gel in the cavity of the gel network [36]. Additionally, the filling effect of CDF due to its insoluble characteristics may also enhance the gel strength of MP emulsion gel according to previous studies.

### 3.4. Water- and Fat-Binding (WFB) Capacity 

WFB is another important functional property of MP gel, indicating the water- and fat-binding capacity of the gel. It can be seen in Figure 3 that the WFB of the emulsion gels increased significantly from 75.38% to 96.28% with the increasing concentration of CDF from 0 to 2%, and the WFB reached the maximum value of 96.28% when 2% CDF was added, which was probably because the much denser gel network structure, which will be mentioned in a later part, can trap water or oil droplets more firmly. This result was similar to the result of Zhuang [10], who found that the WBF of MP gel increased from 74.99% to 89.78% when 2% insoluble dietary fiber was added. 

### 3.5. Low-Field NMR Analysis

The WFB was further evaluated indirectly through the *T*_2_ relaxation time, which can be used to evaluate the mobility of water molecules with different states in the gel system. *T*_2_ relaxation time is characterized by three relaxation components based on the mobility of water molecules (*T*_2b_, *T*_21_, and *T*_22_). Generally, water with a short relaxation time is more tightly bound to macromolecules than that with a long relaxation time [37]. Figure 4A,B show the *T*_2_ relaxation time and peak area of MP gel at different CDF concentrations. As shown in Figure 4A, there were four characteristic peaks in the transverse relaxation curve, protein-associated water (*T*_2b_), immobilized water (*T*_21_), and free water (*T*_22_), respectively [38]. *T*_21_ exhibited two different forms of trapped water (*T*_21a_ and *T*_21b_). For the water in the gel network, *T*_21a_ was much tighter than *T*_21b_ [39]. With the addition of CDF, *T*_2b_ did not change significantly, while *T*_21_ decreased significantly. Notably, when the CDF reached 2%, *T*_21a_ and *T*_21b_ decreased from 21.54 ms and 231.01 ms to 12.33 ms and 114.98 ms, respectively. Similarly, *T*_22_ had a significant reduction from 3274.55 ms to 1629.75 ms, indicating gradually tighter-bound water with MP gel. It is evident from Figure 4B that *PT*_21_ gradually increased from 92.59% (without CDF) to 95.14% (2% CDF) and *PT*_22_ gradually decreased from 1.90% (without CDF) to 0.38% (2% CDF), indicating that more free water was converted to immobilized water. As will be mentioned later, the addition of CDF led to the increase in the surface hydrophobicity and reactive sulfhydryl content of MP. Thus, a much denser gel network was formed to combine water. Moreover, Bertram [40] found that *PT*_21_ was positively correlated with WFB, so the increase in *PT*_21_ was consistent with the results of WFB.

### 3.6. Rheological Properties 

In this study, the effects of angular frequency and temperature on the G’ and G” of MP-CDF emulsion gel were studied. G’ was used to describe the elastic property of gel [41], and gel with a higher value of G’ had better gel elasticity. The effect of changes in angular frequency on G’ and G” can be described by the four most common states: a strong gel, a weak gel, a concentrated solution, and a dilute solution [42]. When G’ was higher than G”, a more flexible gel structure was formed. But a dilute solution is the opposite; G” is always higher than G’. For the weak gel, G’ is higher than G”, and the two lines are almost parallel, while for concentrated solution, the curves of G’ and G” intersect in angular frequency scanning mode [43]. Figure 5 showed the changes in G’ and G” of the composite gel with different concentrations of CDF under angular frequency sweep (Figure 5A) and temperature sweep (Figure 5B).

As shown in Figure 5A, with increase in angular frequency, both G’ and G” increased and had the same trend. Moreover, over the entire angular frequency range, G’ was larger than G”, and the two curves nearly paralleled at the angular frequency range, indicating the formation of an ordered elastic gel. In addition, Figure 5A showed that the MP emulsion had higher values of G′ and G″ with the addition of CDF, which had the maximum value when 2% CDF was added. The results showed that the gel quality was improved with the increasing concentration of CDF. Zhao [44] reported that higher G′ than G″ for the emulsion indicated the formation of an ordered elastic gel structure. 

Next, the temperature-sweep-based rheological behavior of MP after the addition of CDF was evaluated. It can be seen from Figure 5B that the G’ went through three stages during the heating process. The G’ of MP gradually increased at the beginning of heating, indicating the binding of the myosin head and the initial formation of gels. The peak of G’ at 45 °C was thought to be caused by the dimerization of the myosin head. In the next stage, G’ fell sharply to a minimum value at approximately 55 °C. This phenomenon was caused by the unfolding of the myosin tail and the loosening of the binding of the myosin head, which led to the destruction of the protein network [45]. In the subsequent heating process, G’ rose sharply, demonstrating the formation of the gel structure and that the protein was completely denatured. The G’ curve showed the crosslinking and aggregation of proteins, which can be converted from the viscous MP solution to a three-dimensional gel during the heating process. Moreover, the addition of CDF significantly increased the G’ of MP, and the G’ was positively correlated with the concentration of CDF. This was probably because the addition of CDF promoted the intermolecular crosslinking and led to the filling effect due to its insoluble characteristics; then, the structure tended to be compact during the formation of the gel. Chin et al. [46] found that the addition of dietary fiber increased the G’ of the MP system, which was consistent with the results in this study. 

### 3.7. Cryo-SEM Analysis

Cryo-SEM, another variant of traditional SEM, requires freezing the sample prior to imaging and allows the three-dimensional structure of a water-containing sample to be observed under low temperatures and high vacuum conditions [47]. Figure 6 shows the microstructure of the MP emulsion gel with different CDF concentrations. Compared with the control group, the three-dimensional gel network structure of the groups with the addition of CDF was denser and the pores were smaller, which was probably due to the filling effect of CDF, as mentioned above. Zhuang et al. [3] showed that the pores in the gel matrix could be water channels that negatively affect the gel, thereby reducing the water holding capacity of gel. In addition, the particle size of oil droplets gradually decreased with the increasing content of CDF, and there was an interfacial film on the surface of the oil droplets when 2% CDF was added. However, the uneven distribution of the oil droplets was caused by the fact that MP gels were cut into a certain size before freezing. The results showed that CDF could promote the formation of the dense three-dimensional gel network of MP, which provided a basis for improving the structure of the MP emulsion gel. This result was consistent with the results of WFB and gel strength.

### 3.8. Secondary Structure of MP

Circular dichroism (CD) spectroscopy is widely used to determine the change in secondary structure for proteins. The variation in *α*-helix, *β*-sheet, *β*-turn, and random coil proportions were used to explain the change in protein characters. Figure 7 demonstrated the effect of CDF on the secondary structure of MP. It can be seen from Figure 7A that the spectrum exhibited two minima at 208 nm and 222 nm, implying the change in α-helical and secondary structure [48]. Compared with the control, other treatment groups showed a significantly changed secondary structure of MP. As shown in Figure 7B, the *α*-helix content of the groups with the addition of CDF was significantly lower than that of the control MP group, from 18.83% (0% CDF) to 15.51% (2% CDF), and the content of the *β*-sheet increased from 27.70% to 35.22%. These results suggest that CDF promoted the conversion of the *α*-helix to *β*-sheet. A previous study reported that the increase in *β*-sheet promoted the exposure of hydrophobic groups, leading to the increase in surface hydrophobicity [49]. This was consistent with the surface hydrophobicity results shown below. In addition, the temperature for gel formation was also affected by the content of the *β*-sheet, which in turn had an effect on the G’ value of MP. The result was in agreement with the results of Zhuang [13], who reported that MP mixed with SDF had a high content of *β*-sheet, thus improving G′ of the proteins at 80 °C. Liu [50] also reported that the unfolding of *α*-helix and the formation of *β*-sheet favored the gelation of proteins. The results demonstrated that CDF could change the secondary structure of MP by breaking the hydrogen bonding of *α*-helix and promoting the expansion of *α*-helix to form a *β*-sheet structure. 

### 3.9. Surface Hydrophobicity

Surface hydrophobicity is one of the commonly used indices to evaluate a change in protein structure. The increase in surface hydrophobicity means the unfolding of the protein structure and the exposure of hydrophobic amino acids [50]. BPB (bromophenol blue) can be combined with the hydrophobic point of proteins to characterize the surface hydrophobicity of proteins [51]. As shown in Figure 8, adding CDF had a significantly effect on the surface hydrophobicity of MP (*p* < 0.05). Compared with the control, the BPB content increased from 10.18 μg to 22.41 μg with the increase in CDF concentration. The enhanced hydrophobic effect indicated that CDF could promote the expansion of the MP head and exposed more hydrophobic groups inside the protein. As stated by Cao [52], higher surface hydrophobicity led to a decrease in interfacial tension and an increase in EAI. The result was consistent with the emulsification activity. Additionally, MP aggregation was heat induced by exposing hydrophobic groups, so more hydrophobic groups led to the formation of a much denser three-dimensional network structure, thus improving WFB during the subsequent crosslinking heating process.

### 3.10. Reactive Sulfhydryl 

The change in the SH group can reflect a change in tertiary and quaternary structure. R-SH is a mercaptan group that is exposed to the protein surface [53]. Most of them are distributed in the head of myosin, and play an important role in maintaining the tertiary and quaternary structure of proteins [54]. When the structure of myosin changed, the free sulfhydryl groups in myosin were exposed and oxidized to form disulfide bonds. However, due to the polyhydroxy structure, DF can reduce S-S bonds, thus increasing the content of R-SH and exposing more R-SH in proteins. The content of R-SH (mol/10^4^ g) in MP at different CDF concentrations is shown in Figure 9. It can be seen that the content of R-SH increased significantly with the increasing content of CDF, and reached 1.86 mol/10^4^ g and 2.07 mol/10^4^ g, respectively, at 0% and 2.0% CDF. It was found that CDF caused partial unfolding of the protein structure and exposed the sulfhydryl group buried in the protein, leading to the increase in the sulfhydryl group. The change in R-SH content had the same trend as that of surface hydrophobicity. Kim [55] concluded that the exposed hydrophobic groups led to the formation of a dense three-dimensional gel network through hydrophobic interaction. Thus, the emulsion gel properties of MP with CDF could be improved.

## 4. Conclusions

The addition of dietary fiber from chickpeas effectively improved the gel ability of MP emulsion. CDF with concentrations lower than 2% absorbed part of the water and oil in the system and interacted with the proteins under heating conditions, thus affecting the molecular structure and gel properties of the MP. With the increasing CDF concentration, the interaction between MP and CDF promoted the conversion of more *α*-helix to *β*-sheet, exposing more sulfhydryl groups and hydrophobic groups, thus affecting the microstructure, gel strength, and WFB of the final MP gel. Additionally, low-field NMR analysis proved that the addition of CDF increased the content of trapped water, while decreasing the free water content of the MP gel, which further indicated the gradually enhanced water-holding capacity of the MP gel. In summary, the addition of CDF effectively improved the emulsion gel properties of MP by changing the protein structure, and showed great potential to improve the quality of emulsified meat products.

## Figures and Tables

**Figure 1 foods-12-02597-f001:**
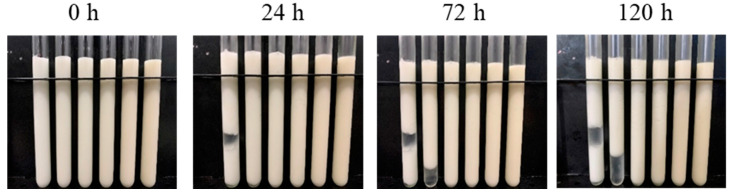
Storage stability photos of MP-CDF emulsions. The concentrations of chickpea dietary fiber (CDF) in emulsions in test tube were 0, 0.4, 0.8, 1.2, 1.6, and 2.0%, from left to right. Appearance images correspond to emulsions after storage for 0, 24, 72, and 120 h, respectively.

**Figure 2 foods-12-02597-f002:**
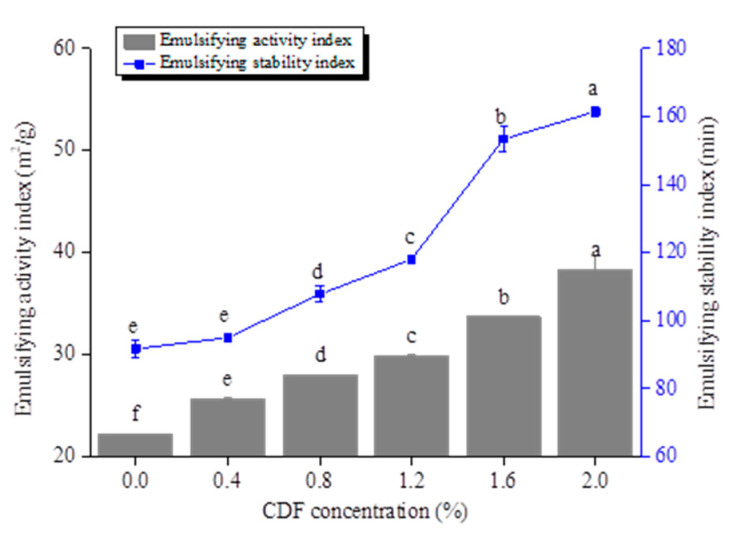
Effect of different CDF concentrations on the emulsifying activity index (EAI) and emulsifying stability index (ESI) of MP. a–f: Different letters above standard deviation bar indicate significant differences among the means (*p* < 0.05).

**Figure 3 foods-12-02597-f003:**
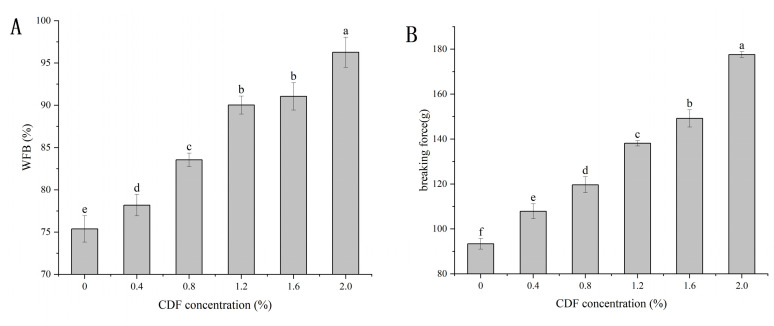
Effect of different concentrations of CDF on the WFB (**A**) and breaking force (**B**) of MP emulsion gel. a–f: Different letters above standard deviation bar indicate significant differences among the means (*p* < 0.05).

**Figure 4 foods-12-02597-f004:**
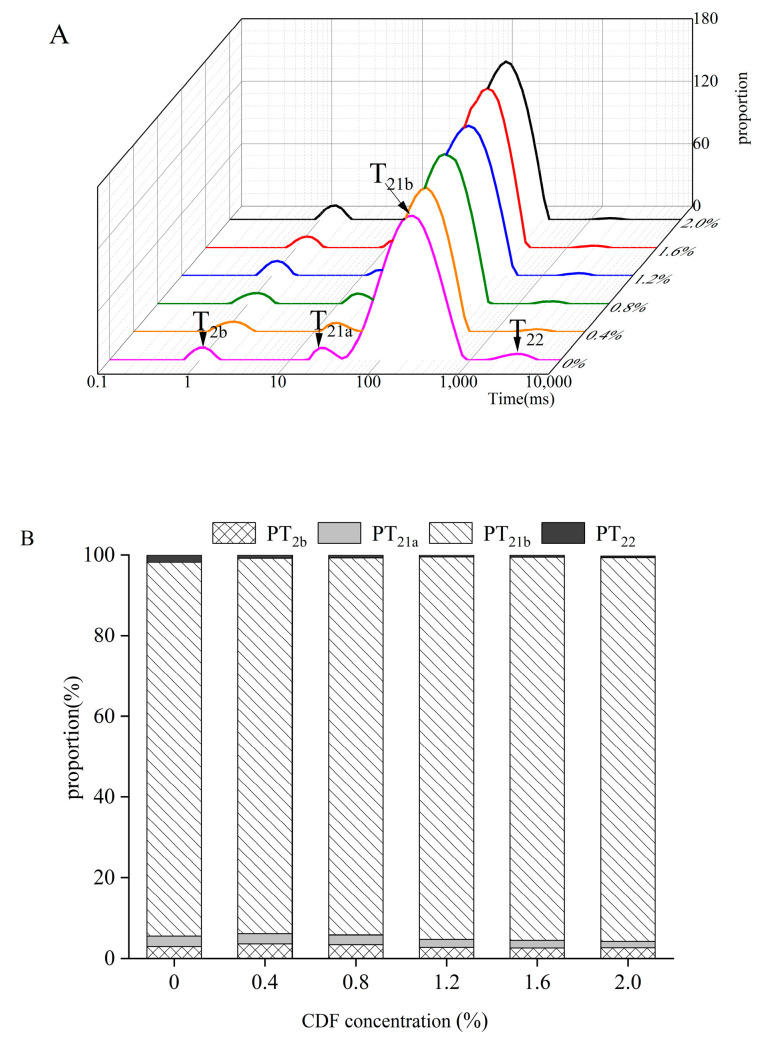
Effect of different CDF concentrations on water distribution of MP emulsion gel. (**A**) represents distributions of T_2_ relaxation times and (**B**) represents proportions of different water in MP-CDF emulsion gels.

**Figure 5 foods-12-02597-f005:**
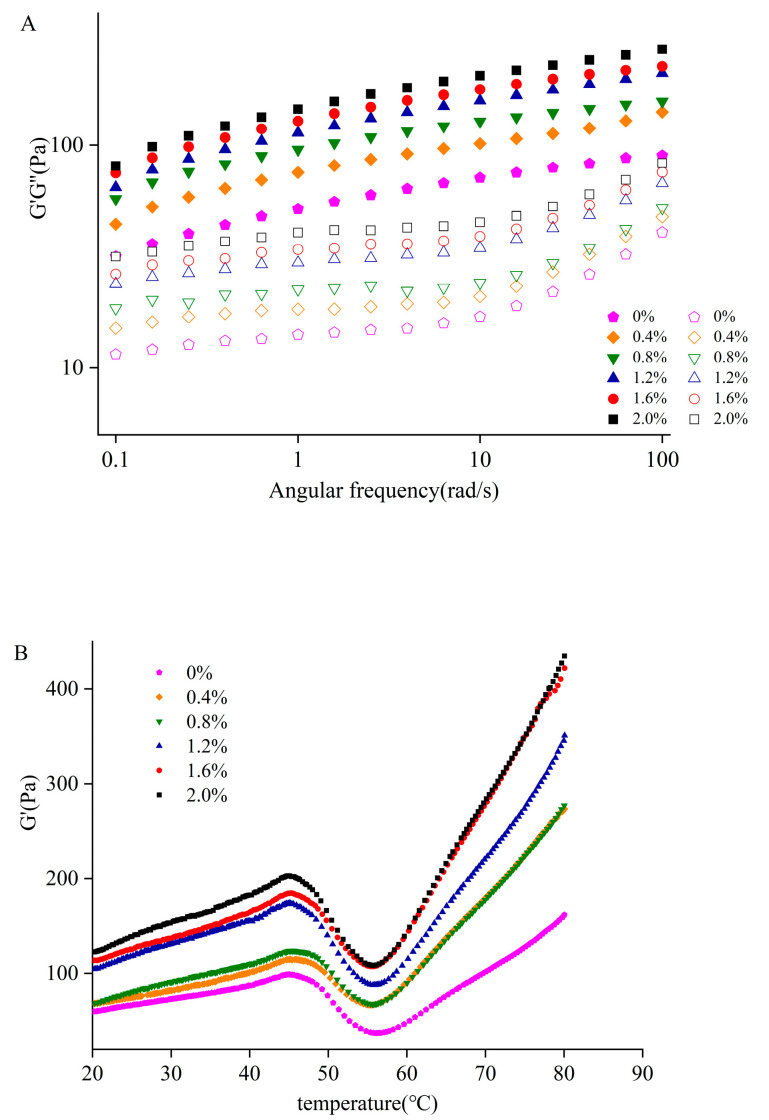
Changes in the storage modulus (G′) and loss modulus (G″) of MP-CDF emulsions. (**A**) represents frequency sweep of emulsions and (**B**) represent changes in G′ of emulsions during heating from 20 °C to 80 °C.

**Figure 6 foods-12-02597-f006:**
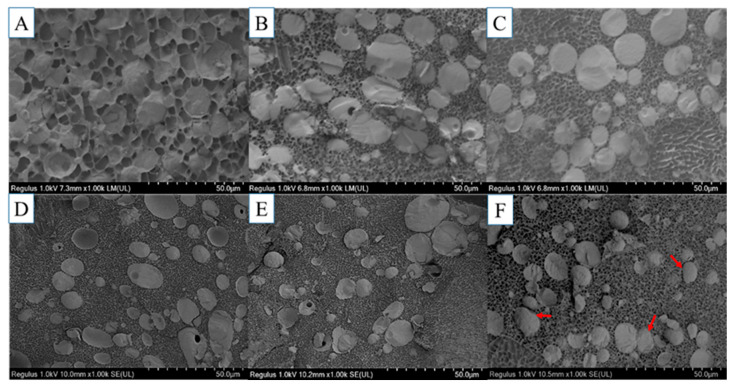
Cryo-SEM micrographs of MP emulsion gel with different CDF concentrations (1000× magnification). (**A**): 0%, (**B**): 0.4%, (**C**): 0.8%, (**D**): 1.2%, (**E**): 1.6%, and (**F**): 2.0%.

**Figure 7 foods-12-02597-f007:**
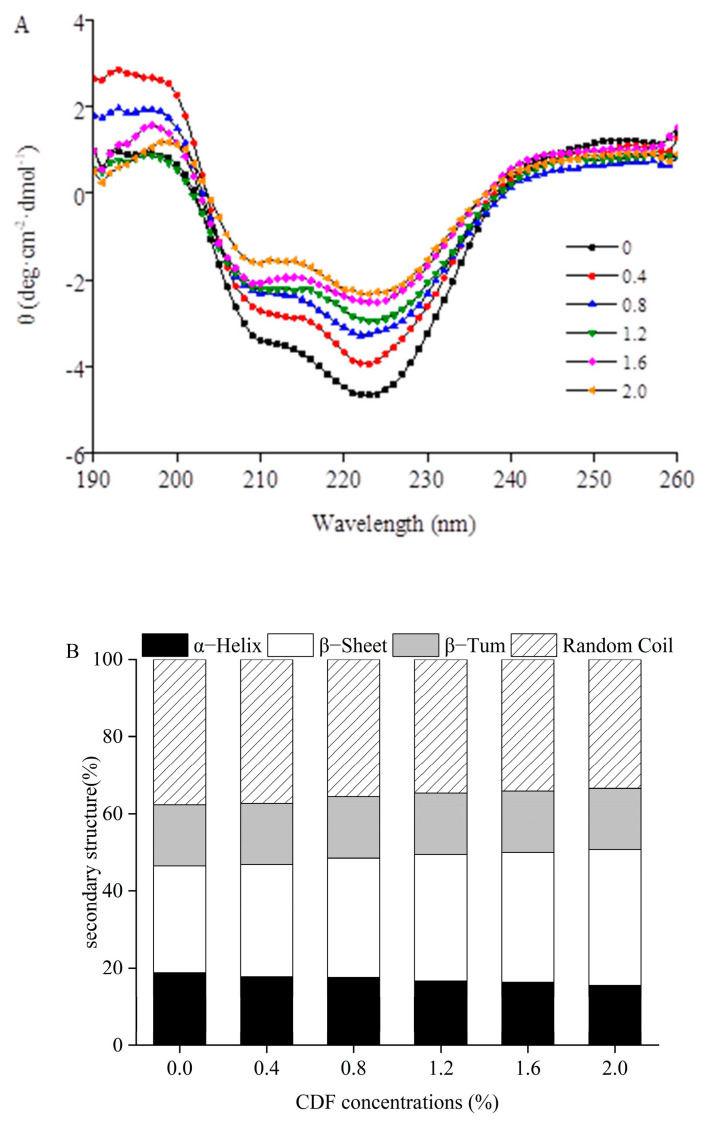
Circular dichroism (CD) spectra (**A**) and secondary structure (**B**) of MP at different CDF concentrations.

**Figure 8 foods-12-02597-f008:**
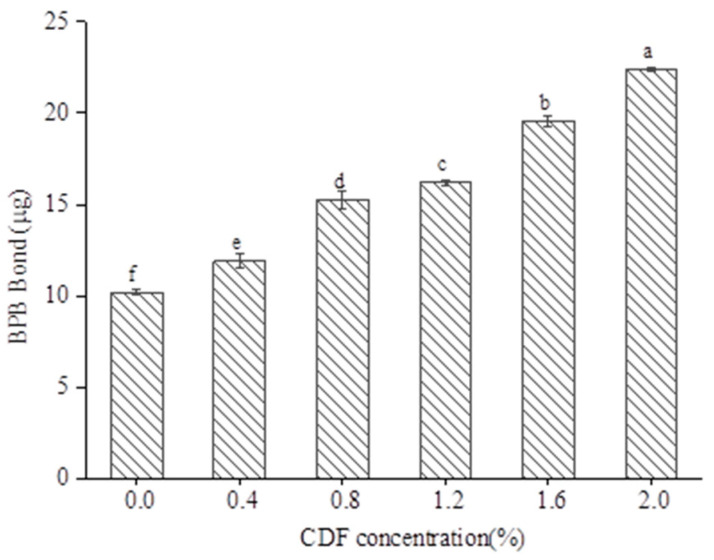
Effect of different CDF concentrations on the surface hydrophobicity of MP. a–f: Different letters above standard deviation bar indicate significant difference (*p* < 0.05).

**Figure 9 foods-12-02597-f009:**
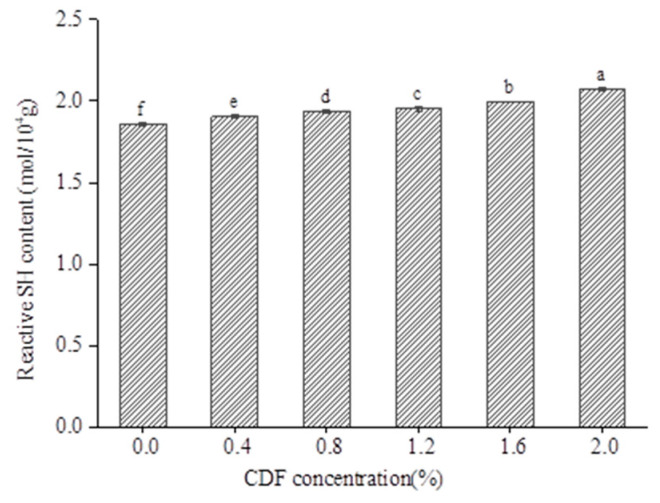
Effect of different CDF concentrations on the reactive sulfhydryl (R-SH) content of MP. a–f: Different letters above standard deviation bar indicate significant difference (*p* < 0.05).

## Data Availability

The datasets generated for this study are available on request from the corresponding author.

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
