# Peer review of "Effect of Chickpea Dietary Fiber on the Emulsion Gel Properties of Pork Myofibrillar Protein"

_foods, 2023, doi:10.3390/foods12132597_

Round 1

Reviewer 1 Report

1.       Title. Please specify the source of myofibrillar protein.

2.       Line 15. For the first-time use, CDF's full name is required. Also, please check the remaining Abstract for "EAI", "ESI", "WFB" and other terms.

3.       Line 15. % of what?

4.       Line 19. What was improved, please specify?

5.       Line 23-24. Describe the "change" in detail. Was it elevated or lowered? How does it affect the gel's properties?

6.       Line 38. The definition of "emulsion gels" is required.

7.       Line 46-48. Please provide some examples of additives that have been used to enhance emulsion gel.

8.       Line 60. Please specify the source of MP.

9.       Line 67. ….a small amount of fat (5.5%~6.9%) Is it really small? It is almost 7%.

10.   Line 70. What type of MP?

11.   Line 81. How long did it take for frozen storage?

12.   Line 81. Italicize the scientific name.

13.   Line 88-89. To verify that the myofibrillar proteins in the sample are solubilized, please calculate and provide the ionic strength of the extraction buffer.

14.   Line 96-105. Why only DF? Why not make a comparison to its flour?

15.   Line 104-105. Please give the yield of the CDF.

16.   Line 131. Why 80 °C for 30 min?

17.   Line 132. Please specify the room temperature?

18.   Line 133-139. Why the total fluid was measured? Why not “fat binding” and “water binding” abilities.

19.   Line 147. Please give the way to calculate “gel strength”.

20.   Line 207. Please add a brief introduction why this parameter was determined.

21.   The authors' discussion on the storage stability of emulsions was found in section 3.1. I did not observe the specifics of the study's methodology. Therefore, please add the method's detail.

22.   In Fig. 2, the EAI and ESI tended to rise as CDF increased. Why was the upper limit of 2% chosen? Why not attempt with greater concentration? Also, the same trend can be seen in Fig. 3.

23.   Line 250. Is the term “aggregation” optimal? It should be “gel-forming ability”

24.   Line 402-418. Why the reactive RH increased with increasing CDF content? How about the disulfide bond formation during gelation of MF (see line 406-407)?

25.   Line 425. The "Proper Concentration" from this study should be stated. Since the authors only allowed for 2% CDF in the test, it is not precisely a realistically adequate level of concentration. Increasing the CDF may further enhance the gel. So, this is the weak point.

Please double check English.

Author Response

Thanks for your suggestion for the improvement of this paper, I have made a change to this paper accordingly and marked the change in red in the manuscript.

  1. Please specify the source of myofibrillar protein.

Response:  I have specified the source of myofibrillar protein in the title

  1. Line 15. For the first-time use, CDF's full name is required. Also, please check the remaining Abstract for "EAI", "ESI", "WFB" and other terms.

Response:  I have used the full name of these terms.

  1. Line 15. % of what?

Response:  This is the concentration of the CDF, which means the concentration of the whole system.

  1. Line 19. What was improved, please specify?

Response:  I have specified the terms in Line 21.

  1. Line 23-24. Describe the "change" in detail. Was it elevated or lowered? How does it affect the gel's properties?

Response:  I have described the "change" in detail in line 25.

  1. Line 38. The definition of "emulsion gels" is required.

Response:  I have added the definition of "emulsion gels" in Line 39.

  1. Line 46-48. Please provide some examples of additives that have been used to enhance emulsion gel.

Response:  I have provided two examples of additives in line 50 and line51.

  1. Line 60. Please specify the source of MP.

Response:  I have specified the source of MP in Line 64.

  1. Line 67. ….a small amount of fat (5.5%~6.9%) Is it really small? It is almost 7%.

Response:  I have changed this term in Line 71. 

  1. Line 70. What type of MP?

Response:  I have mentioned the type of MP in Line 75. 

  1. Line 81. How long did it take for frozen storage?

Response:  Usually the the meat was frozen stored for less than one week.

  1. Line 81. Italicize the scientific name.

Response:  I have Italicized the scientific name in Line 83.

  1. Line 88-89. To verify that the myofibrillar proteins in the sample are solubilized, please calculate and provide the ionic strength of the extraction buffer.

Response:  Because the concentration of NaCl significantly influenced the solubility of myofibrillar proteins, so I have provided the concentration of NaCl in Line 93 and Line 97.

  1. Line 96-105. Why only DF? Why not make a comparison to its flour?

Response:  Because the flour contained oil and starch, so the components were complicated. While DF from the flour was more healthy, so DF was used.

  1. Line 104-105. Please give the yield of the CDF.

Response:  I have added the yield of the CD in Line 112.

  1. Line 131. Why 80 °C for 30 min?

Response:  This was according to a previous research and I have added a reference in Line 25.

  1. Line 132. Please specify the room temperature?

Response:  I have specified the room temperature in Line 142.

  1. Line 133-139. Why the total fluid was measured? Why not “fat binding” and “water binding” abilities.

Response:  We calculated the summary of fat binding and water binding ability, so the total fluid was measured.

  1. Line 147. Please give the way to calculate “gel strength”.

Response:  We used the data “Breaking force” to represent “gel strength”, and the breaking force was directly obtained from data.

  1. Line 207. Please add a brief introduction why this parameter was determined.

Response:  I have added a a brief introduction in Line219.

  1. The authors' discussion on the storage stability of emulsions was found in section 3.1. I did not observe the specifics of the study's methodology. Therefore, please add the method's detail.

Response: I have added the methodology in Line 222.

  1. In Fig. 2, the EAI and ESI tended to rise as CDF increased. Why was the upper limit of 2% chosen? Why not attempt with greater concentration? Also, the same trend can be seen in Fig. 3.

 Response:  We aimed to investigate the law that CDF affect the the EAI and ESI of MP, so we chose different concentration of CDF. 

Line 250.  Is the term “aggregation” optimal? It should be “gel-forming ability”

Response:  I have changed the term “aggregation” into  “gel-forming ability” in Line266.

  1. Line 402-418. Why the reactive RH increased with increasing CDF content? How about the disulfide bond formation during g

elation of MF (see line 406-407)?

Response:  This was probably because CDF contained a small amount of polyphenol, then disulfide bond can be reduced to reactive RH, so the the reactive RH increased with increasing CDF content.  As the content of reactive RH increased, more disulfide bond will be formed during the gelation of MP.

  1. Line 425. The "Proper Concentration" from this study should be stated. Since the authors only allowed for 2% CDF in the test, it is not precisely a realistically adequate level of concentration. Increasing the CDF may further enhance the gel. So, this is the weak point.

Response:  Yes, you are right, we have changed this term in Line 444.

Reviewer 2 Report

The paper entitled “Effect of chickpea dietary fiber on the emulsion gel properties of myofibrillar protein” deals with the possibility of incorporating fibers of chickpea in order to improve protein properties. Various concentrations of dietary fiber were used to this purpose and several functional properties of the myofibrilar protein were followed.

The approached subject is interesting especially in the actual context of the continuous efforts of developing new ways of improving the food quality.

The aims of the study are clearly expressed.

The experimental program is described in such manner that, with slight amendments, it can be easily applied.

The obtained results are concise, clearly presented and discussed.

Conclusions are drawn according to the obtained data.

The authors will find bellow some minor corrections and adjustments that should be addressed.

 -          The inclusion of the abbreviations meaning at their first appearance in the text (e.g. in the Abstract section there are abbreviations for which the explanation is available later in the text) is necessary. 

-          There are phrases requiring attention (e.g. “While MP emulsion gel is mainly dependent on the formation of three-dimensional network structure of MP in aqueous phase, and its structural characteristics are key factors affecting the properties of emulsion gel.”; “Evaluating the improved functional functional properties of MP by addition of CDF has important guiding significance for the formulation of meat products.”; “MP was diluted with 0.6 M NaCl buffer to ensure the final protein concentration was 30 mg/mL”. etc.)

-          Latin names should be written with italics letters (e.g. Cicer arietinum Linn) in all the manuscript.

-          Providers of all chemicals used for the experiments should be added.

-          Providers for all equipment and specific working conditions are mandatory (e.g. freeze-drying, grinder and sieving equipment and conditions used in section 2.3 are not given etc.)

-          The defatted chickpea flour was obtained in laboratory or purchased as such? Authors should add the method used or the provider.

-          It is recommended to ad a brief explanation on why using defatted chickpea.

-          Several different concentrations of chickpea dietary fiber were tested (0.4 %, 0.8 %, 1.2 %, 1.6 % and 2 %). A justification of this selection should be included. Values higher than 2 % were considered? What would be the effect of using more than 2 % of CDF?

Minor editing of English language required

Author Response

Response to reviewer 2:

Thanks for your suggestion for the improvement of this paper, I have made a change to this paper accordingly and marked the change in red in the manuscript.

The inclusion of the abbreviations meaning at their first appearance in the text (e.g. in the Abstract section there are abbreviations for which the explanation is available later in the text) is necessary. 

Response:  I have given more explanation for the abbreviations in the abstract.

-          There are phrases requiring attention (e.g. “While MP emulsion gel is mainly dependent on the formation of three-dimensional network structure of MP in aqueous phase, and its structural characteristics are key factors affecting the properties of emulsion gel.”; “Evaluating the improved functional functional properties of MP by addition of CDF has important guiding significance for the formulation of meat products.”; “MP was diluted with 0.6 M NaCl buffer to ensure the final protein concentration was 30 mg/mL”. etc.)

-          Latin names should be written with italics letters (e.g. Cicer arietinum Linn) in all the manuscript.

   Response:  I have written the Latin names with italics letters (e.g. Cicer arietinum Linn) in all the manuscript.

-          Providers of all chemicals used for the experiments should be added.

Response:  I have added the the providers of all chemicals used for the experiments in Section 2.1.

-          Providers for all equipment and specific working conditions are mandatory (e.g. freeze-drying, grinder and sieving equipment and conditions used in section 2.3 are not given etc.)

Response:  I have added all providers for all equipment and specific working conditions in section 2.3. 

-          The defatted chickpea flour was obtained in laboratory or purchased as such? Authors should add the method used or the provider.

     Response:  The defatted chickpea flour was obtained in laboratory, I have added the details in Line 102.

-          It is recommended to add a brief explanation on why using defatted chickpea.

Response:  I have added a brief explanation on why using defatted chickpea in Line102.

-          Several different concentrations of chickpea dietary fiber were tested (0.4 %, 0.8 %, 1.2 %, 1.6 % and 2 %). A justification of this selection should be included. Values higher than 2 % were considered? What would be the effect of using more than 2 % of CDF?

   Response:  We aimed to investigate the law that CDF affect the emulsion gel properties of MP, so we chose different concentration of CDF. As we found that the the emulsion gel properties of MP were greatly improved with increasing content of CDF, so adding appropriate content of CDF can meet the requirements of meat processing. So we do not consider the values higher than 2 %.

Round 2

Reviewer 1 Report

1. Please rechek the use of abbreviations in Line 17 and 18. It should read as " emulsifying activity index (EAI) and emulsifying stability index) (ESI) of MP increased with increasing content of CDF. Moreover, the water and fat binding capacity (WFB)...."

2. Line 81. "Porcine" It no needs to italicize.

Author Response

Response to reviewers:

Thank you for the suggestion and comments, I have made change to the manuscript accordingly and marked the change in red.

Line 83: as suggested by the reviewers the Latin name "Cicer arietinum Linn" has to be written in Italic.

Response: I have written "Cicer arietinum Linn" in Italic in Line 85.
Line 91: as required by the reviewer 2 the provider of instruments (the centrifuge in this case) has to be reported.

Response: I have provided the provider of instruments in Line95, Lin96 and Line141.   

Line 92: "… the pellets were re-suspended with a homogenizer…" which type of homogenizer?

Response: I have provided the type of homogenizer in Lin 96.  
Line119-120; "… The fresh emulsion was immediately collected from the bottom and …" which bottom? Why is it collected form the bottom?

Response: The emulsion was collected from the bottom of the tube for preparing emulsion. I have made a change in Line124. This is according to previous research and I have added the corresponding reference.  
"…. After shocking for 30 s, the absorbance was determined…" maybe "… after vortexing …" as said in the original reference or "after mixing " could be clearer

Response: I have made this change in Line125.
Line 132: which homogenizer?

Response: I have made this clear in Line142.
Line 140: when determining Water and fat binding (WFB) capacity, after centrifugation I suppose that supernatant is removed weighting samples… this should be said

Response: I have added this detail in Line151.
Line 151-152: as required by reviewer 1, a definition of gel strength, the procedure to estimate it is necessary. The performed test gives, as a result, a plot of force as a function of displacement (or velocity). How did the authors identify the gel strength? Moreover, during discussion, gel strength is not cited whereas the authors discuss about the Breaking force. Are these parameters the same? In this case only one name (either gel strength or breaking force) has to be used to avoid confusion.

Response: I have used breaking force in all the manuscript and added more details in Line156-Line165.
Line 163: The exact rheometer model should be given, according to the manufacturer website it could be Discovery HR30, or Discovery HR20, or Discovery HR10. Please specify the exact model. Moreover I suppose that "American" should be "USA"

Response: I have made this clear in Line176.
Line 168; Why 1% strain amplitude was chosen to perform dynamic tests? Are the authors sure that this value can guarantee linear viscoelastic conditions?

Response: 1% strain amplitude was chosen based on pre-experiment and this value can guarantee linear viscoelastic conditions.

Line 168: replace "… The energy storage modulus (G') and loss modulus (G")…" with "… The dynamic storage modulus (G') and loss modulus (G")…"

Response: I have made a change accordingly in Line 181.
Line 171: I suppose that "silica gel oil" should be "silicon oil"

Response: I have made a change in Line 184.

Line 174: " The cooling rate was 10 °C/min." No results of cooling tests are shown. If this is the cooling rate used at the end of the test only before removing the sample the information is not necessary and it can be removed form the manuscript.

Response: I have removed this information.
Line 212-216: according to reviewer 1 the procedure for storage stability has to be given in "materials and methods" section not here. In "results and discussion" section results of storage stability are discussed

Response: I have added this part in Line 135-138.
Line 326: G' and G" are not "directly proportional" to CDF concentration, no evidence of this direct proportionality is given (maybe a non linear trend can be seen in graphs)

Response: I have deleted this sentence.
